# Comprehensive Treatment Algorithms of the Swiss Peritoneal Cancer Group for Peritoneal Cancer of Gastrointestinal Origin

**DOI:** 10.3390/cancers14174275

**Published:** 2022-09-01

**Authors:** Michel Adamina, Maxime Warlaumont, Martin D. Berger, Silvio Däster, Raphaël Delaloye, Antonia Digklia, Beat Gloor, Ralph Fritsch, Dieter Koeberle, Thibaud Koessler, Kuno Lehmann, Phaedra Müller, Ralph Peterli, Frédéric Ris, Thomas Steffen, Christian Stefan Weisshaupt, Martin Hübner

**Affiliations:** 1Klinik für Viszeral- und Thoraxchirurgie, Kantonsspital Winterthur, 8401 Winterthur, Switzerland; 2Faculty of Medicine, University of Basel, 4056 Basel, Switzerland; 3Chirurgie Digestive et Cancérologique, CHU de Lille, CH de Cambrai, 59000 Lille, France; 4Department of Visceral Surgery, Lausanne University Hospital CHUV, University of Lausanne, 1011 Lausanne, Switzerland; 5Department of Medical Oncology, Inselspital, Bern University Hospital, University of Bern, 3010 Bern, Switzerland; 6Clarunis, Department of Visceral Surgery, University Centre for Gastrointestinal and Liver Diseases, St. Claraspital and University Hospital Basel, 4031 Basel, Switzerland; 7Department of Medical Oncology, University Hospital Basel, 4031 Basel, Switzerland; 8Department of Oncology, Lausanne University Hospital CHUV, University of Lausanne, 1011 Lausanne, Switzerland; 9Department of Visceral Surgery and Medicine, Inselspital, University Bern, 3010 Bern, Switzerland; 10Department of Medical Oncology and Hematology, University Hospital Zurich, 8091 Zurich, Switzerland; 11Department of Medical Oncology and Hematology, St. Claraspital, 4002 Basel, Switzerland; 12Faculty of Medicine, University of Bern, 3012 Bern, Switzerland; 13Department of Oncology, Geneva University Hospital, 1205 Geneva, Switzerland; 14Department of Surgery and Transplantation, University Hospital Zurich, 8091 Zurich, Switzerland; 15Division of Digestive Surgery, University Hospitals of Geneva, 1205 Geneva, Switzerland; 16Klinik für Allgemein-, Viszeral-, Endokrine und Transplantationschirurgie, Kantonsspital St. Gallen, 9000 St. Gallen, Switzerland; 17Division of Medical Oncology and Hematology, Kantonsspital St. Gallen, 9000 St. Gallen, Switzerland

**Keywords:** peritoneal carcinomatosis, cytoreductive surgery, hyperthermic intraperitoneal chemotherapy, HIPEC, gastrointestinal cancer, pressurized intraperitoneal chemotherapy, PIPAC

## Abstract

**Simple Summary:**

Peritoneal cancer is best addressed by a multimodal treatment approach, including cytoreductive surgery, systemic immunochemotherapy, and intraperitoneal chemotherapy (HIPEC, PIPAC). The present Swiss Peritoneal Cancer Group comprehensive treatment algorithms offer consensus guidance for interdisciplinary care of patients with peritoneal cancer from pseudomyxoma peritonei, peritoneal mesothelioma, gastric, and colorectal origin. They include straight complete surgical resection, multimodal neoadjuvant treatment, HIPEC and PIPAC, and referral to palliative care. These state-of-the-art algorithms have been endorsed by all Swiss clinicians routinely involved in the multimodal care of patients with peritoneal cancer of gastrointestinal origin.

**Abstract:**

Peritoneal cancer (PC) is a dire finding, yet in selected patients, long-term survival is possible. Complete cytoreductive surgery (CRS) together with combination immunochemotherapy is essential to achieve cure. Hyperthermic intraperitoneal chemotherapy (HIPEC) and pressurized intraperitoneal aerosol chemotherapy (PIPAC) are increasingly added to the multimodal treatment. The Swiss Peritoneal Cancer Group (SPCG) is an interdisciplinary group of expert clinicians. It has developed comprehensive treatment algorithms for patients with PC from pseudomyxoma peritonei, peritoneal mesothelioma, gastric, and colorectal origin. They include multimodal neoadjuvant treatment, surgical resection, and palliative care. The indication for and results of CRS HIPEC and PIPAC are discussed in light of the current literature. Institutional volume and clinical expertise required to achieve best outcomes are underlined, while inclusion of patients considered for CRS HIPEC and PIPAC in a clinical registry is strongly advised. The present recommendations are in line with current international guidelines and provide the first comprehensive treatment proposal for patients with PC including intraperitoneal chemotherapy. The SPCG comprehensive treatment algorithms provide evidence-based guidance for the multimodal care of patients with PC of gastrointestinal origin that were endorsed by all Swiss clinicians routinely involved in the multimodal care of these challenging patients.

## 1. Introduction

Peritoneal cancer (PC) comprises a heterogeneous group of primary peritoneal tumors and metastatic disease from various origins. Common characteristics are late diagnosis, a limited response to systemic therapy, and hence, a dismal prognosis [1,2]. New treatment modalities and multimodal strategies have improved prognosis considerably over the last decade and a cure has become possible for selected patients. Complete cytoreductive surgery (CRS) with or without heated intraperitoneal chemotherapy (HIPEC) provides the best outcomes for most entities but entails risks for postoperative complications and a long recovery period [3,4,5,6]. While the intent of CRS is to remove all visible disease, intraperitoneal (ip) chemotherapy addresses microscopic residual disease, so as to maximize the benefit of an extensive surgical resection. PC occurs frequently in colorectal and gastric cancers. Indeed, up to 25% of relapsing colorectal cancer patients develop metastatic disease restricted to the peritoneum, while around 8% of patients present with isolated PC at primary resection [7,8,9]. In gastric cancer, isolated PC is found in 5–20% of patients who undergo surgical exploration for potentially curative resection [5]. In infrequent cancers such as pseudomyxoma peritonei and peritoneal mesothelioma, CRS HIPEC is a standard of care, whenever a patient is fit for major surgery [4,6,10,11,12,13,14,15,16]. Thus, in expert centers providing CRS HIPEC, cure rates above 80% are reported for pseudomyxoma peritonei [6,10,17,18,19].

Conversely, the debate is ongoing for colorectal and gastric cancers that represent the vast majority of patients affected by PC, who are largely treated with palliative systemic therapy. Indeed, it is estimated that more than 90% of these patients only receive a systemic chemotherapy combined with a biological agent, whereas about 5% may be treated in a multimodal approach, including CRS HIPEC [20]. This is unfortunate and unfair to our patients as the evidence in favor of cytoreductive surgery is strong, with a controversy limited to the additional value of HIPEC in addition to a complete cytoreduction. Several international registries as well as phase 2 and phase 3 randomized controlled trials have underlined the benefit of a curative approach taking advantage of CRS HIPEC in PC of gastric and colorectal origin [21,22,23,24,25,26,27], with a single, although most recent, randomized controlled trial not concluding in favor of HIPEC in colorectal cancer [22]. Beyond PC originating from gastrointestinal cancers, another large European multicentric randomized controlled trial demonstrated a remarkable overall survival benefit of one year in women with stage III ovarian cancer who benefited from HIPEC in addition to CRS [28]. Nonetheless, morbidity and resource utilization remain of concern when considering the true value of CRS HIPEC. Indeed, the nationwide Dutch CRS HIPEC prospective registry reported a mortality rate of 3% for a major morbidity of 34% and a length of stay of 16 days, which is similar to the clinical outcomes of most major oncological procedures. Importantly, numerous studies have reported that a maximal treatment including the addition of HIPEC to CRS was worth the trouble in the patients’ perspective, with acceptable to high quality of life related within a few months after CRS HIPEC [29,30,31,32,33,34,35].

Recently, the surgical administration of intraperitoneal chemotherapy evolved further to include pressurized intraperitoneal aerosol chemotherapy (PIPAC) administered laparoscopically [36]. Phase I and II studies have reported safety and efficacy of PIPAC for a variety of cancers [37,38,39], including low morbidity and preservation or improvement of quality of life [40,41,42,43]. Moreover, iterative laparoscopy when performing PIPAC allows for repeat biopsy and objective assessment of tumor regression with a validated grading system to guide multimodal treatment [44]. Hence, both CRS HIPEC and PIPAC now belong to the advanced armentarium of an efficient multimodal approach to peritoneal cancer.

Today, clinicians and tumor boards that advise treatment for patients with PC are confronted with a large and dynamic body of literature in a rapidly evolving, highly specialized clinical context. Yet, RCT or prospective comparative studies are rare for most clinical situations. The aim of the present comprehensive treatment algorithms by the Swiss Peritoneal Cancer Group (SPCG) is therefore to propose a standardized and pragmatic approach for multimodal treatment of PC of gastrointestinal origin.

## 2. Materials and Methods

The SPCG was founded in 2012 as a working group within the society of Swiss Visceral Surgeons. Since inception, a medical oncologist with an academic practice has been a member of its executive board and instrumental in securing a true interdisciplinary vision. This practice consensus was elaborated along a modified Delphi process as follows:

### 2.1. Review of the Literature and Drafting of Comprehensive Treatment Algorithms

Pertinent literature and current guidelines on the treatment of PC of gastrointestinal origin were scrutinized by two authors from the core team from Lausanne. The literature search focused on pseudomyxoma peritonei, peritoneal mesothelioma, and PC of gastric and colorectal origin. Empirical studies including patients undergoing CRS with or without ip chemotherapy (HIPEC, PIPAC) and published in English, German, or French were included. References of the included studies were checked for additional missed articles. The best available evidence was collated giving priority to RCTs, comparative prospective studies, large-scale retrospective studies, and clinical registry data. Well-performed systematic reviews and meta-analyses were considered and searched for further references. Opinion statements, editorials, grey literature, and expert opinion were not considered in agreement with current recommendations [45]. Four working algorithms with accompanying text were drafted.

### 2.2. Internal Validation of the Lake Geneva Algorithms

The 4 treatment algorithms and their accompanying text were submitted for internal review to surgical and medical oncologists involved in the treatment of PC patients at the university hospitals of Lausanne and Geneva. Substantial modifications of the algorithms were discussed and implemented during a 4-month working period including 3 face to face Delphi rounds.

### 2.3. External Validation of the SPCG Algorithms

The Lake Geneva algorithms were presented to the board of the SPCG and revised. The literature was cross-checked and updated, and the algorithms were thoroughly discussed until a consensus was reached in a further Delphi round. The SPCG algorithms were complemented by in-depth discussion and alignment to nationwide clinical practice in Swiss expert centers until a final consensus was reached and unanimously endorsed by the board of the SPCG and the lead PC clinician of 7 surgical oncology and 7 medical oncology departments. The SPCG comprehensive treatment algorithms for PC of gastrointestinal origin were presented to the international audience and faculty of the SPCG symposium and validated during the general assembly of the SPCG on 3 September 2021.

The SPCG treatment algorithms are endorsed by all Swiss institutions routinely offering extensive cytoreductive surgery with multivisceral resection and HIPEC or PIPAC, including 7 surgical oncology and 7 medical oncology departments. They are intended as representative guidance for interdisciplinary tumor boards and clinicians taking care of patients with PC and not as formal guidelines. This practice consensus intends to standardize multimodal care of patients with PC in Switzerland and promote nationwide inclusion in the prospective SPCG registry.

PC of gynecologic origin is not covered in the present treatment algorithms, yet a similar approach is currently underway by the respective working group of the SPCG.

## 3. Results

### 3.1. Pseudomyxoma Peritonei

Pseudomyxoma peritonei is a rare cancer originating from a ruptured low grade mucinous neoplasia of the appendix (LAMN) with an estimated incidence of 2 to 4 per million [46,47,48]. Rarely, it may also arise from the colon, the pancreas, or the ovary [49]. It is often an incidental finding either upon radiologic examination or when performing an abdominal procedure, including appendectomy for suspected appendicitis. Left untreated, mucinous tumor cells accumulate by gravity at the sites of peritoneal fluid uptake, in the pelvis, along the rima coli and the greater and lesser omentum, and under the diaphragm. Deep infiltration of organs is not happening in the early course of the disease and the small bowel tends to be spared. Accumulation of mucus causes slowly progressive abdominal distension and organ dysfunction, including intestinal obstruction, cachexia, and ultimately, death. Metastatic disease outside the abdomen is uncommon.

Surgical treatment is the only curative option, yet the cure necessitates a complete cytoreduction. Cytoreduction typically includes a right colectomy, radical omentectomy, cholecystectomy, pelvic peritonectomy, bilateral parietal, and diaphragmatic peritonectomy, and in women, hysterectomy with salpingo-oophorectomy. Additional peritonectomy (liver capsule, mesentery), splenectomy, and bowel resection are performed when infiltration is suspected. Complete cytoreduction is essential and it proceeds irrespective of the peritoneal cancer index, with the remaining small bowel length and preservation of vital organ function being the only technical limitation. Extensive surgery is challenging for the patient and the surgeon alike, frequently lasting 6 to 12 h. Yet, when supplemented with HIPEC, a long-term cure rate above 80% can be achieved [6,10,17,18,19,50].

Neoadjuvant treatment is not generally recommended, owing to a poor response rate. Adjuvant chemotherapy is, however, advisable in selected cases with high recurrence risk, e.g., incomplete resection, high grade pseudomyxoma, and increasing peritoneal cancer index [51]. Measurement of the tumor markers CEA, CA 19-9, and CA 125 is often elevated and if so, it may prove useful in the follow-up.

About a quarter of the patients treated with optimal CRS HIPEC recur [51,52]. Of those qualifying for a repeat CRS HIPEC, long-term survival and morbidity is similar to the figures observed in primary CRS HIPEC patients [51,52,53]. A large registry study totalizing 1924 patients with pseudomyxoma peritonei treated until December 2017 compared patients treated with cytoreductive surgery alone to the conventional CRS HIPEC regimen: it showed better overall survival and reasonable morbidity in patients treated with CRS HIPEC. Hence, the SPCG recommends CRS HIPEC for patients deemed fit for extensive surgery who present with a pseudomyxoma peritonei. The treatment algorithm proposed by the SPCG is illustrated in Figure 1. These recommendations are consistent with the current guidelines of the Peritoneal Surface Oncology Group International (PSOGI) and with most national guidelines [54].

Performing a staging laparoscopy allows one to individualize treatment according to the extent and histology at hand [55]: in the presence of localized mucin without epithelial cells, simple laparoscopic follow-up without CRS HIPEC may be offered; conversely, high-grade with signet-ring histology tumors may receive neoadjuvant systemic treatment, followed by CRS HIPEC. Notably, there is no threshold peritoneal cancer index for pseudomyxoma peritonei other than technical resectability and fitness of the patient for a major resection.

Pressurized intraperitoneal aerosol chemotherapy [56,57] is the latest addition to the interdisciplinary armentarium for the treatment of PC. PIPAC may also be considered in patients with pseudomyxoma peritonei, in particular, for unresectable cases [56]. However, clinical experience in this context is still limited and no firm recommendation for PIPAC can be made at present.

### 3.2. Peritoneal Mesothelioma

Malignant mesothelioma is affecting the serosal membranes of the pleura, peritoneum, pericardium, or tunica vaginalis testis. The peritoneum is the second most frequent site following the pleura. It is a rare and highly lethal cancer with an incidence of about 0.7 to 3 per million people [58,59]. It has been linked to asbestos exposure, yet the association is weaker than with pleural mesothelioma and no sex predominance exists. Patients present with abdominal distension/pain, weight loss/anorexia, while bowel obstruction is a manifestation of advanced disease. Diagnosis relies on percutaneous core needle biopsy or explorative laparoscopy with biopsy, rather than on cytological examination of serosal effusion or fine needle biopsy [12]. It is advisable to place all laparoscopic trocars along the midline, so as to allow for a straightforward resection at laparotomy and minimize port site metastasis.

The conventional classification distinguishes two main forms: diffuse malignant peritoneal mesothelioma (DMPM) and borderline forms, which include multicystic peritoneal mesothelioma (MCPM) and well-differentiated papillary peritoneal mesothelioma (WDPPM). DMPM itself comprises three histological subtypes (epithelioid, sarcomatoid, or biphasic) (12). Expert pathology review is recommended to differentiate histological subtypes and estimate clinical course. Measurement of the tumor marker CA 125 is advised in addition to cross-sectional imaging and laparoscopy [12].

The two main forms show different behavior: the rarer well-differentiated papillary mesothelioma progresses slowly and metastasizes late; the more frequent diffuse malignant peritoneal mesothelioma shows an aggressive behavior and responds poorly to chemotherapy [60]. In patients of both subtypes fit for surgery, CRS HIPEC is a curative approach with no exclusion per se because of a high peritoneal cancer index. Watchful waiting is an alternative approach in the frail patient with a well-differentiated papillary peritoneal mesothelioma with treatment escalation when progressive disease is detected. In the presence of an epithelioid subtype, exploratory laparoscopy is advised to assess the true extent of disease, as no imaging modality is able to reliably assess resectability of PC of any origin. Some centers advise one to take advantage of exploratory laparoscopy to initiate intraperitoneal chemotherapy with administration of a first PIPAC. Indeed, neoadjuvant PIPAC has been shown to be effective and of low morbidity, in particular, when performed ahead of any major surgery or associated with neoadjuvant systemic chemotherapy [61,62]. Neoadjuvant therapy is offered whenever initial evaluation reveals a high risk peritoneal mesothelioma (peritoneal cancer index > 17, Ki-67 > 9%, sarcomatoid histology, nodal positive status, incomplete resection CC score >1), and/or borderline resectability. Following neoadjuvant systemic chemotherapy, re-evaluation is performed, including another explorative laparoscopy—which again may be supplemented by a PIPAC with the purpose to improve resectability [63]. Once a patient is deemed resectable and fit for surgery, CRS HIPEC is offered.

In recurrent peritoneal mesothelioma following curative CRS HIPEC, another CRS HIPEC is rarely offered but palliative systemic chemotherapy, which again may be supplemented by palliative PIPAC. Similarly, non-resectable mesothelioma is treated by systemic chemotherapy and/or PIPAC [56,60,64,65]. Of note, new data regarding the use of immune checkpoint inhibitors in the palliative setting have to be taken into consideration, especially regarding the sarcomatoid histology, but are not yet mature enough to be integrated in this algorithm. The treatment algorithm proposed by the SPCG is illustrated in Figure 2. These recommendations are consistent with the 2021 published guidelines of the Peritoneal Surface Oncology Group International [12] and with the Chicago consensus on peritoneal surface malignancies [16].

### 3.3. Gastric Cancer

In Western countries, gastric cancer often presents in symptomatic, late stages of disease and it is hence often incurable. It is, however, a common cancer ranking 13^th^, with an incidence of 8.1 per 100,000 people in Europe (5.1/100,000 in Switzerland) and a cure rate of 37.5% [66].

The European Society of Medical Oncology recommends multimodal treatment for patients with stage ≥ IB resectable gastric and gastroesophageal junction cancers, including laparoscopic staging, neoadjuvant chemotherapy, surgery, and adjuvant chemotherapy [67,68]. Tumor staging includes computed tomography (CT) and frequently endoscopic ultrasound, which may provide more accurate locoregional staging than CT, as well as biopsy. Laparoscopy completes tumor staging and provides direct visualization of peritoneal surfaces and local lymph nodes, as well as biopsy of any suspicious lesions. Often, PC is diagnosed during laparoscopic staging and/or peritoneal washing. Again, no diagnostic imaging can rule out PC so that laparoscopic staging is advised in most curative situations. Indeed, PC is detected in 15–53% of gastric cancers treated with curative intent [69,70]. Measurement of the tumor markers CA 19-9 and CEA have a prognostic value and can be used in the follow-up, while HER2 and microsatellite status can be sought after as a biomarker with therapeutic implications. Last, multiplex profiling has shown that CDH1 and TAF1 mutations, 6q loss and chr19 gain were seen more frequently in PC of gastric origin, while more aggressive PC phenotypes were emerging with increased mutations in TP53, CDH1, TAF1, KMT2C, and chromosomal instability [71]. However, the impact of molecular screening has not yet translated into clinical care.

In the presence of PC, whenever the patient is deemed resectable and presents a peritoneal cancer index not greater than 6, we advise for standard neoadjuvant chemotherapy, which may be supplemented by 2 concomitant PIPAC to optimize resectability and cure rate. Indeed, PIPAC performed in the context of unresectable GC achieved a pathologic response in more than 60% and allowed conversion to resectability in up to 14% of the patients [61,63,72,73]. The addition of PIPAC to systemic neoadjuvant chemotherapy further allows for a dynamic laparoscopic response evaluation, including iterative biopsies and tumor regression grading to inform oncologic treatment and resectability of the primary cancer and PC. CRS and HIPEC, including D2 lymphadenectomy and peritonectomy then follow, completed by adjuvant chemotherapy as mentioned in European guidelines [67,68]. Prospective cohort series and randomized trials in Asian patients have shown a survival advantage for CRS HIPEC in gastric cancer patients presenting with PC [74,75]. An updated large multicentric French cohort study included 277 patients and reported a consistent benefit of CRS HIPEC over CRS alone in multivariable analysis, irrespective of histology. Median OS was of 16.7 vs. 11.3 months (poorly cohesive carcinoma), respectively, 34.5 vs. 14.3 months (non-poorly cohesive carcinoma). Peritoneal cancer index below 7 (poorly cohesive carcinoma), respectively, 13 (non-poorly cohesive carcinoma) were predictive of OS [5,76]. Similar results were reported from multicentric Italian and German cohort studies [26,77]. The results of the German GASTRIPEC RCT are expected to be published shortly following presentation at ESMO 2021. The trial was powered to detect a benefit in overall survival in 180 patients. It was stopped after 105 patients or less than 60% of its planned accrual due to slow recruitment and 55 patients who stopped treatment before CRS (disease progression or death). Similar morbidity and overall survival were reported, together with a clinically and statistically significant benefit in progression-free (3.5 months) and distant metastasis-free survival [78].

Conversely, when a patient is deemed unresectable or progresses during neoadjuvant chemotherapy, palliative second line treatment is offered, which can also be supplemented with PIPAC for optimal response and quality of life with conversion to resectability reported in few cases [38,79]. PIPAC is also a valuable option for control of ascites and other symptoms in patients refractory to systemic treatment [38,40,57,64,79]. Of note, the role of immune checkpoint inhibitors as a new standard of care for gastric cancer [80] in the metastatic setting has yet to be defined in the context of cytoreductive surgery.

The treatment algorithm proposed by the SPCG is illustrated in Figure 3. It is important to note that supplementing perioperative chemotherapy and surgery with CRS HIPEC allows for a 5-year overall survival of up to 27% in metastatic patients who would most likely have died earlier otherwise [5,26,27,75,81,82,83,84]. In addition, neoadjuvant PIPAC has been successfully assessed in many specialized centers [61,62,85] and it is currently investigated in a dedicated trial [86].

### 3.4. Colorectal Cancer

Colorectal cancer is the third most common cancer with an incidence of 30.4 per 100,000 people in Europe (22.3/100,000 in Switzerland) and a cure rate of 59.5%% [66]. At the time of diagnosis, up to 10% of patients with colorectal cancer have synchronous PC and more than half of the patients with recurrent disease will present with metachronous PC [87,88,89]. Of the 3 histological subtypes of colorectal cancer (adenocarcinoma (85–90%), mucinous adenocarcinoma (10–15%), signet ring cell carcinoma (1%)), PC occurs predominantly in mucinous and signet ring histologies [90], while adenocarcinoma preferably metastasize to the liver. When PC is left to conventional palliative chemotherapy, median overall survival does not exceed 16 months [91]. Recent developments in the management of advanced colorectal cancer include total neoadjuvant chemotherapy, targeted therapies, and refinements in the indication and practice of CRS HIPEC. Yet, the fate of patients with PC from colorectal cancer is grim, as no systemic chemotherapy including biologic agents can offer any prospect of a cure [1,91,92], as opposed to complete cytoreductive surgery—possibly supplemented with intraperitoneal chemotherapy. A large body of literature, including multicentric prospective clinical registries, suggests a survival benefit when complete cytoreductive surgery is achieved and HIPEC is performed [4,18,24,93,94,95], which has led to inclusion of CRS HIPEC in several national guidelines. As conventional imaging performs poorly for the early detection of PC, diagnostic laparoscopy may be offered in colorectal cancer patients at high recurrence risk within 6–12 months of the primary bowel resection or when the tumor marker CEA rises during follow-up with no obvious metastatic disease in radiological staging.

Two randomized controlled trials have assessed CRS HIPEC in PC of colorectal origin. The first and smaller Dutch trial (n = 105) compared standard oncologic resection and adjuvant systemic chemotherapy to CRS HIPEC and adjuvant systemic chemotherapy. It showed a 5-month progression-free survival benefit for CRS HIPEC performed with Mitomycin C and a significant improvement in disease-free survival from 12.6 months to 22.2 months in the CRS HIPEC group. Median survival was 48 months with a 5-year survival rate of 45% in patients who achieved a complete cytoreduction supplemented by HIPEC [21]. The second, larger (n = 265) and recent French trial (PRODIGE-7) randomized patients peroperitavely who had just undergone complete cytoreduction and had neoadjuvant systemic chemotherapy to receive HIPEC or no further treatment. It found no survival benefit for CRS HIPEC performed with high-dose oxaliplatin ip and 5-FU/leucovorin iv, except for patients with a peritoneal cancer index of 11–15 in a post-hoc analysis [22]. The latter point is consistent with a prior finding suggesting that patients with a peritoneal cancer index of up to 16 and/or limited small bowel involvement benefit most from CRS HIPEC [96,97,98,99]. Much has been written on the strength and weakness of the PRODIGE-7 trial, which was first presented in June 2018 at the American Society of Clinical Oncology meeting and included contemporary systemic chemotherapy and targeted therapy. Complete cytoreduction was confirmed as the cornerstone of a curative approach in PC with an overall survival of 41.7 months for CRS HIPEC and of 41.2 months for CRS alone, and a recurrence-free survival of 13.1 vs. 11.1 months, respectively. While survival and 30-day morbidity did not differ significantly, 60-day morbidity was significantly higher in the CRS HIPEC arm compared to CRS alone (24.1% vs. 13.6%). Potential causes for the late morbidity were heavily preoperative systemic treatment for 6 months and an aggressive course of ip/iv intraoperative chemotherapy with maximal dosage of ip oxaliplatin (460 mg/m^2^) and hyperthermia (43 °C) that may ultimately have made CRS HIPEC patients more vulnerable to late medical complications [100]. Indeed, routine preoperative administration of a 6-month course of oxaliplatin-based systemic chemotherapy is not common practice across countries, nor is the aggressive HIPEC regimen of PRODIGE-7. A Dutch randomized trial (CAIRO6) currently near completion may shed light soon on the benefit of perioperative systemic chemotherapy in patients who undergo CRS HIPEC.

Signet ring cell histology is an independent unfavorable prognostic factor, regardless of the treatment approach. Data from the nationwide Dutch cancer registry have shown that the relative survival gain of CRS HIPEC is comparable for adenocarcinoma, mucinous adenocarcinoma, and signet ring histology, while systemic therapy improved survival in all histological subtypes [101]. Since patients with signet ring histology are often younger, CRS HIPEC may thus be offered in highly selected patients with an absolute survival gain of up to 10.9 months when compared to systemic therapy, respectively 18 months when compared to supportive care only [101].

From a molecular standpoint, BRAF mutation is overrepresented and found in up to 26% of patients with colorectal PC [102], while KRAS and BRAF mutations negatively impact survival independently of the use of current targeted therapy (95). However, a national prospective cohort of Norvegian patients treated with CRS HIPEC between 2005 and 2015 did not see any differences in survival according to KRAS of BRAF mutations. Interestingly, patients who presented with a BRAF mutation and microsatellite instability had significantly better survival [103]. In a further analysis of 505 patients who underwent CRS HIPEC at 4 European centres, KRAS mutations and to a lesser extent positive nodal stage were independent predictors of peritoneal recurrence following CRS HIPEC [104].

The Biological score of peritoneal metastasis (BIOSCOPE) was recently proposed to help patient assessment and selection. It takes into account peritoneal cancer index, nodal status, differentiation grading, and KRAS/BRAF mutations and allows categorization of patients into 4 survival groups with a prediction performance of 0.70 (development/validation area under the curve). It underlined that RAS/RAF mutations impair survival after CRS HIPEC, independently of the use of current targeted therapy [105]. Tumor biology is a key element to factor in when choosing intensity and sequence of therapies.

As of Summer 2021, German [106], French (https://www.snfge.org/content/4-cancer-colorectal-metastatique (accessed on 18 July 2022)), British [107], American [108], and Canadian [109] practice guidelines stated that CRS HIPEC can be considered in experienced centers for selected patients. Selection amounts to fitness for major surgery and completeness of cytoreduction, both of which are mandatory. In addition, the extension of PC of colorectal origin matters beyond mere resectability with a threshold peritoneal cancer index of 20 and below most commonly reported. Indeed, many expert centers today set the bar for CRS HIPEC at a peritoneal cancer index of 16 [96,97,98,99,110,111] to maximize cure, while avoiding large resection of small bowel to prevent a short bowel syndrome. Peritoneal recurrence after CRS HIPEC is common with a median time to recurrence of 33 months [112,113,114]. Repeat CRS HIPEC can be offered in highly selected patients with similar results as for a primary procedure, once extraperitoneal metastases have been excluded [53,115].

The treatment algorithm proposed by the SPCG is illustrated in Figure 4. Upfront diagnostic laparoscopy is recommended to assess the diagnosis of PC and resectability, thus overcoming the limitations and low reliability of PC imaging [116]. It may include a first PIPAC to initiate treatment of PC [61]. In the presence of resectable PC, systemic chemotherapy is performed, followed by repeat staging and CRS HIPEC, pending objective response being confirmed. Additional systemic chemotherapy follows CRS HIPEC, whenever a full regimen has not been given prior to CRS HIPEC.

When, however, diagnostic laparoscopy reveals a borderline resectable situation, in particular, a peritoneal cancer index greater than 15 in a patient otherwise fit for surgery [117], a maximal approach includes neoadjuvant systemic therapy combined with 2 further PIPAC. Indeed, a published prospective series and clinical experience of Swiss centers speak for the safety and effectiveness of joint systemic and PIPAC chemotherapy as a means to optimize a response and select the patient with a chance for a cure [61,63]. Repeat staging follows, and final selection either for a curative approach including CRS HIPEC as above or for a palliative approach is done. The SPCG has drafted a Swiss multicentric randomized controlled trial to quantify the effect of PIPAC when added to standard second line systemic therapy in colorectal PC and currently is seeking funding for it.

Lastly, when a patient is deemed unresectable, palliative systemic chemotherapy may be enhanced by iterative PIPAC, including ascites control and relief. The development of extraperitoneal metastatic disease or tumor progression during neoadjuvant systemic chemotherapy is further grounds to opt for a palliative approach.

## 4. Discussion

Peritoneal carcinomatosis affects a significant number of patients with cancers of gastrointestinal origin. Treatment options are evolving rapidly, and the prospect of a cure is realistic for selected patients who are willing and able to undergo a demanding multimodal treatment in specialized cancer centers. CRS HIPEC is a key element in this context and a complete cytoreduction is the cornerstone of every curative approach. The addition of intraperitoneal chemotherapy to complete cytoreduction is a logical step, as peritoneal relapses are frequent and driven by non-visible/non-resectable PC against which chemotherapy is best active when administered in situ. Repeat ip chemotherapy has been tested and deemed effective, yet it encompassed prolonged intraabdominal tubing (in- and effluent of chemotherapy) or repeat laparotomies, both of which are prone to morbidity. Iterative PIPAC combines the ease of a minimally invasive approach, including repeat biopsy to objectively monitor tumor response, and the advantage of repeat ip administration of chemotherapy. More than 215 publications document the effectiveness of PIPAC in the treatment of PC and multiple clinical trials include PIPAC in the multimodal approach to PC [118]. The combination of systemic chemotherapy and PIPAC is a further evolution in the treatment of PC [119] and an integral part of the present SPCG comprehensive treatment algorithms that builds up on multiple current oncologic guidelines. The present SPCG treatment algorithms are endorsed by 7 surgical oncology and 7 medical oncology departments; they reflect the forefront thinking and practice of specialized Swiss centers. They complement existing recommendations by including PIPAC in their standard of care, which is a reimbursed procedure in Switzerland since 2016 [117]. They are intended as an outline for interdisciplinary discussion at tumorboards.

Providing contemporary treatment of advanced malignancies requires specialized knowledge of all involved caregivers, including medical and surgical oncologists, nutritional support, and intensive care physicians. Best practice for perioperative management of patients with CRS HIPEC has been established [120]. Patient care is at its best when clinicians monitor their own outcomes prospectively and participate in translational and clinical research. Hence, the SPCG has initiated in 2021 a dedicated, nationwide prospective clinical registry for all patients considered for CRS HIPEC and/or PIPAC. The SPCG registry allows clinical research and facilitates participation to clinical trials. All Swiss institutions providing CRS HIPEC and PIPAC services are either board members of the SPCG or have confirmed their participation to the SPCG registry, so that a nationwide coverage is warranted. Beyond morbidity, mortality, length of hospital stay, recurrence-free and overall survival, the SPCG registry monitors the rate of complete cytoreduction, the multimodal treatment protocol followed, and the case load of each institution. The Dutch and the British have reported their nationwide experience in standardization and specialization of care in dedicated centers for the treatment of peritoneal surface malignancies, while a minimal case load of 10 CRS HIPEC per surgeon/institution and year has been proposed as a threshold to maintain proficiency [6,121,122]. In our experience, meeting such a threshold requires a catchment area of half a million, whereas a larger referral zone of one million and an institutional volume of 20 CRS HIPEC are advisable. Looking at three metrics from the nationwide Dutch experience (960 CRS HIPEC patients) allows for performance benchmarking: the reported rate of complete cytoreduction was 80% for a median hospital stay of 16 days and a mortality rate of 3%. In terms of survival, median overall survival was 33 months (95% CI 28–38 months) for colorectal cancer and 130 months (95% CI 98–162 months) for pseudomyxoma patients [122,123]. Current nationwide Norwegian data report a median survival of 49 months from the time of CRS HIPEC and a 5-year overall survival of 40.1% for colorectal cancer [103], in concordance with results from British and French tertiary referral centers [22,95]. These outcomes are realistic figures that have been met and exceeded by specialized Swiss centers [104,105].

## 5. Conclusions

The cornerstone of a curative approach in the treatment of PC is completeness of cytoreduction, which requires a high surgical expertise and routine in the care of multivisceral resections. The addition of ip chemotherapy supports completeness of cytoreduction by eliminating unseen tumor clusters prone to relapse. PIPAC is a much needed and validated addition to the treatment of PC: its inclusion to the present recommendations supports a quest for excellence by repeat delivering chemotherapy in situ and providing an objective response grading to guide multimodal treatment. The inclusion of all patients treated with CRS HIPEC and PIPAC in Switzerland in the prospective SPCG registry allows for performance benchmarking and supports clinical research. It is hoped that the present comprehensive treatment algorithms will provide guidance for interdisciplinary discussion at tumorboard and help for optimal and tailored care for patients with PC of gastrointestinal origin.

## Figures and Tables

**Figure 1 cancers-14-04275-f001:**
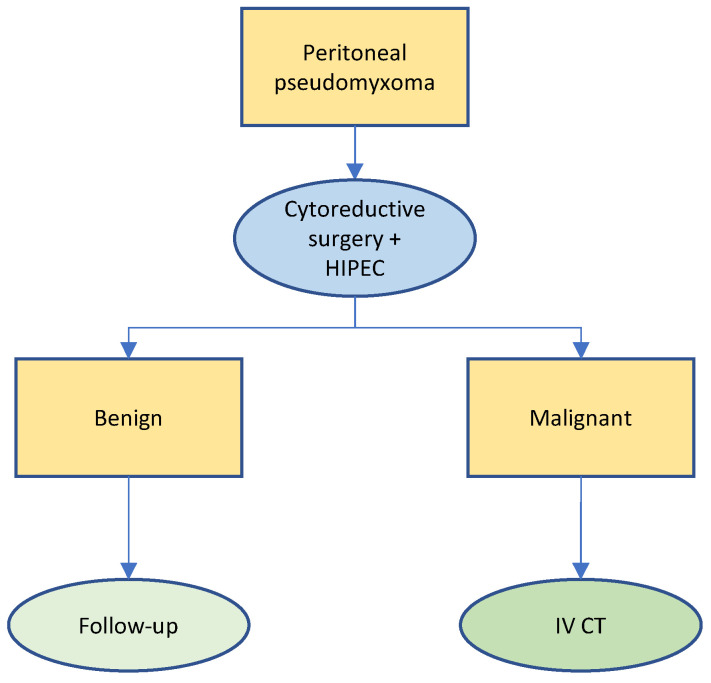
Treatment algorithm for pseudomyxoma peritonei.

**Figure 2 cancers-14-04275-f002:**
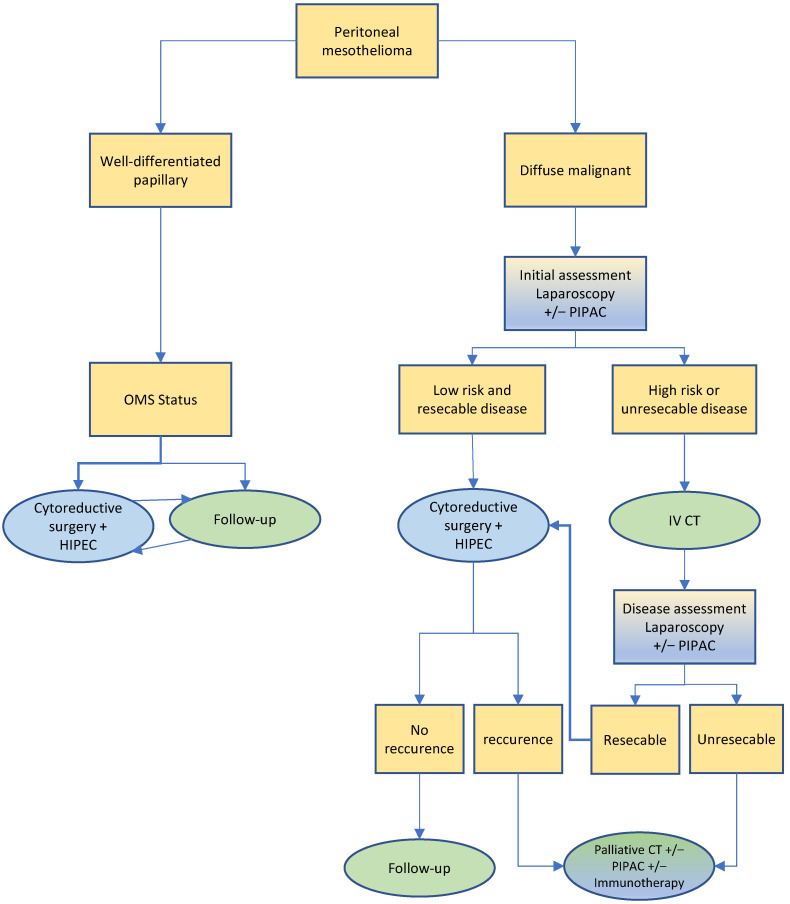
Treatment algorithm for peritoneal mesothelioma.

**Figure 3 cancers-14-04275-f003:**
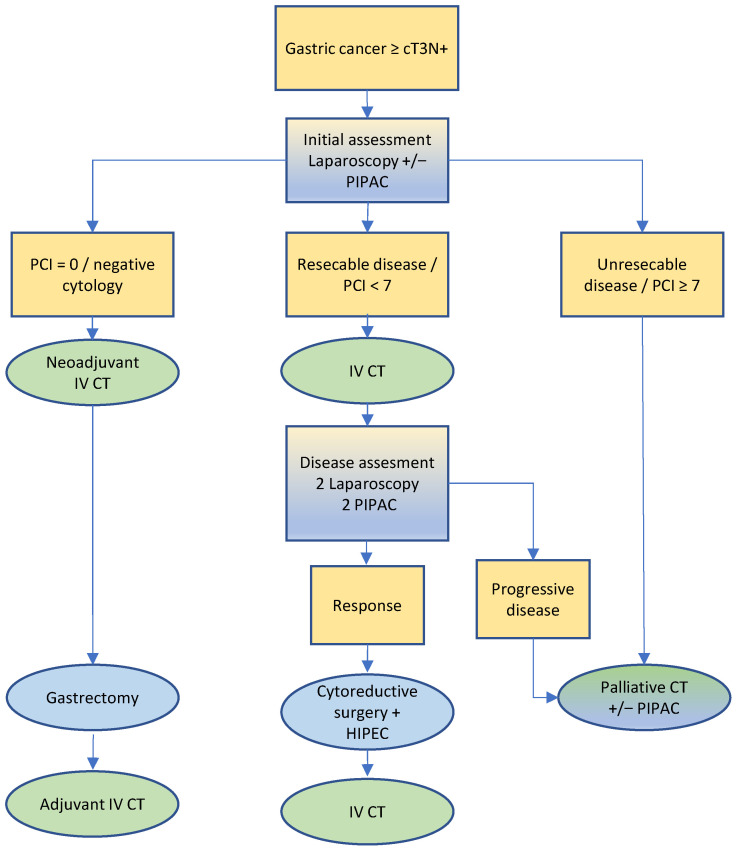
Treatment algorithm for gastric cancer.

**Figure 4 cancers-14-04275-f004:**
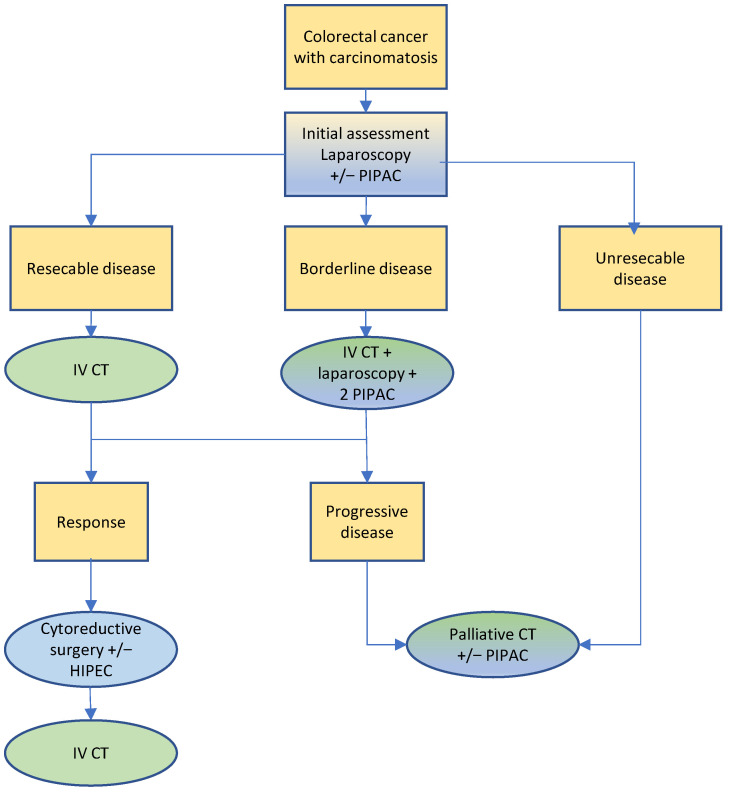
Treatment algorithm for colorectal cancer.

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
