# Peer review of "Comprehensive Treatment Algorithms of the Swiss Peritoneal Cancer Group for Peritoneal Cancer of Gastrointestinal Origin"

_cancers, 2022, doi:10.3390/cancers14174275_

Round 1

Reviewer 1 Report

I would like to start by congratulating our Swiss colleagues for proposing national disease-specific protocols. This means huge documentation and preparation work and a very convincing plea in front of colleagues in order to reach consensus. This attitude can be an inspiration for other groups.

However, I have several comments that do not concern the proposals undertaken by the Swiss group but rather the rationale or the phrasing in the final text. As this article can be an inspiration for other societies, extreme caution to scientific truth is of utmost importance.

Major comments:

1. I am extremely confused about line 294-296. They convey the idea that it is standard for a RESECTABLE PCI of gastric cancer to propose PIPAC. This strategy is under testing by the VerONE trial and is under no circumstances supported by ref 61 that discusses UNRESECTABLE PSM of gastric origin (PCI>8-10). While the Swiss group might have validated the strategy currently tested by VerONE trial, please rephrase the rationale to be consistant with status quo.

2 For line 355, please explain that reference 21 compared CRS+HIPEC vs systemic chemotherapy as the two RCTs in CRC respond to completely different questions. Similarly, please state the comparison arm for ref 22. In my opinion we cannot just suppose that our reader is already informed and the parallel construction induces the wrong idea (positive MMC trial, negative Ox trial). While the latter is true, the first situation vs CRS alone is currently under test by a Spanish RCT.

Minor comments:

1. For reference concerning QoL in resected patients with PSM, please consider replacing ref 34 with Moaven O et al, Ann Surg Oncol, 2020 - health related QoL data coming from a RCT

2. please consider adding a reference for extraappendicular PMP: Delhorme JB et al, Br J Surg, 2018 is one of the potential choices but similar articles might be preferred by the authors

3. line 216: I find the expression "in presence of unresectability" a bit confusing. Maybe "for unresectable cases"?

4. line 253: neoadjuvant chemotherapy is not REQUIRED in patients with PCI>17 but can be proposed. Please rephrase with a softer and more adequate statement

Author Response

Please see the attachment - cover letter and point-by-point response - Thank you.

Reviewer 2 Report

The article is very well written and clear. However, it lacks a chapter on the drugs used according to histology. What drugs are used? Is there a standardization between the different teams in the Swiss Peritoneal Cancer Group? What are their recommendations for the drugs to be used, the dose, the duration of the HIPEC, the temperature,...

Author Response

(The authors gave the same response as above.)

Reviewer 3 Report

I thank you to give me the opportunity to review this interesting article. The paper is very well written. Each etiology is well described in a clear style.

The position of CRS +/- HIPEC and the role of PIPAC are clearly presented  and well explained. the authors did an extensive literature review for a global approach of the topic. Futhermore the authors do not forget to remind context and current discussion goes on around  these IP approaches in some etiology as colorectal peritoneal carcinomatosis.

Although international guidelines have been recently published for primary peritoneal rare diseases (PSOGI, Chicago Group), this article should contribute to disseminate these recommendations locally. And the consensual approach chosen by the authors to elaborate these "helvetic" guidelines (incl. medical oncologists ans surgeons) through discussions with successive levels of validation should expect a hugh acceptance of these by Swiss oncological GI communauty.

Author Response

(The authors gave the same response as above.)
